# Recent Advances on F-Doped Layered Transition Metal Oxides for Sodium Ion Batteries

**DOI:** 10.3390/molecules28248065

**Published:** 2023-12-13

**Authors:** Hao Wang, Lifeng Zhou, Zhenyu Cheng, Liying Liu, Yisong Wang, Tao Du

**Affiliations:** 1State Environmental Protection Key Laboratory of Eco-Industry, School of Metallurgy, Northeastern University, Shenyang 110819, China; 2201664@stu.neu.edu.cn (H.W.);; 2Engineering Research Center of Frontier Technologies for Low-Carbon Steelmaking (Ministry of Education), School of Metallurgy, Northeastern University, Shenyang 110819, China

**Keywords:** sodium-ion batteries, transition metal oxides, F-doping, mechanism, electrochemical performance

## Abstract

With the development of social economy, using lithium-ion batteries in energy storage in industries such as large-scale electrochemical energy storage systems will cause lithium resources to no longer meet demand. As such, sodium ion batteries have become one of the effective alternatives to LIBs. Many attempts have been carried out by researchers to achieve this, among which F-doping is widely used to enhance the electrochemical performance of SIBs. In this paper, we reviewed several types of transition metal oxide cathode materials, and found their electrochemical properties were significantly improved by F-doping. Moreover, the modification mechanism of F-doping has also been summed up. Therefore, the application and commercialization of SIBs in the future is summarized in the ending of the review.

## 1. Introduction

Since the industrial revolution [1], the greenhouse effect and extreme weather events more frequently take place, caused by the increasing CO_2_ emissions from the fossil fuels [2]. As such, we urgently need to find clean renewable energy sources to avoid climate deterioration [3]. However, renewable energy sources such as wind, solar, wave, and geothermal energy are intermittent, unstable, uncontrollable, and have other disadvantages [4]. Aiming to use these energy sources efficiently, large-scale electrochemical energy storage systems (EESSs) are needed to store this energy and convert it into electricity for continuable use [5]. Therefore, the development of advanced energy storage techniques to store intermittent renewable energy has gradually become more important [6]. Owing to the high safety, the high efficiency of conversion, the low cost, and environmental friendliness, rechargeable batteries are regarded as one of the most competent representatives of energy storage and conversion technologies. In addition, the peaks and valleys of the daily electricity consumption usually do not coincide with the peaks and valleys of renewable energy generation, thus storing electricity via batteries can perfectly solve this problem [7].

Since LIBs were discovered in the 1970s and officially commercialized for powering electronic products in 1991 [8], they have achieved great success with great advantages in practical applications, but they are limited by the extremely uneven distribution and the high price of lithium ore [9]. Their electrolytes contain non-aqueous flammable organic solvents that pose a serious fire hazard in the event of thermal runaway [10]. In the 1980s, researchers discovered that Na^+^ could be reversibly (de)inserted from NaCoO_2_ layered oxides. Since then, SIBs have gradually attracted the attention of researchers. Because sodium and lithium are located in the same main group in the periodic table of elements and have similar physical and chemical properties, SIBs and LIBs have similar charging and discharging operating principles [11]. Due to the uniform distribution of sodium resources in the world from sea water, SIBs have the advantages of low cost, abundant element reserves, environmental friendliness, and high safety [12], and have great potential to replace LIBs as a new generation of energy storage devices. So, SIBs have been considered as one of the most promising energy storage systems to substitute for LIBs in the field of large-scale energy storage [13]. However, at the current stage of SIB research, there are many problems that are still restricting their future commercialization application to fully meet the requirement for future EESSs. (1) Lower energy density: the current commercial LIBs energy storage density is much higher than the actual energy storage density of SIBs [14]; (2) Structural damage: due to the larger atomic radii of Na^+^ than that of Li^+^, Na^+^ repeatedly embedding in and out of the cathode material will cause greater damage to the structure of the cathode material, meaning that the conventional cathode for LIBs cannot be directly applied to SIBs [15]; (3) Complex phase transition: the complex outer ring electronic structure and the wide range of valence states of transition metals lead to crystal structure distortion and complex electrochemical behaviors during the charging and discharging process [16]; (4) Instability in air: SIBs normally show poor air stability, which may originate from insertion into the interlayer sites of water or carbon dioxide and the formation of alkaline species such as NaOH or Na_2_CO_3_, leading to surface structure degradation or even failure [17]. Therefore, for the development of the energy industry, new electrode materials with less shortcomings for SIBs should be developed to meet the needs of commercial applications. To achieve high performance for SIBs, the best endeavors have been devoted to exploiting a broad range of new materials, pursuing an outstanding cycling stability, ultra-stable structure, and improving Na^+^ transport kinetics.

It is obvious that the cathode material is the key to the electrochemical performance of SIBs based on the energy storage mechanism [18]. A variety of compounds have been used as cathode materials for SIBs, such as layered transition metal oxides [19], polyanionic compounds [20], Prussian blue analogues [21], and organic materials [22]. Among the leading cathode candidates, layered transition metal oxide cathodes have aroused much interest due to their higher special capacity, faster Na^+^ diffusion rate, and easier industrial synthesis. In recent years, researchers have made great contributions and significant breakthroughs in the optimization of Na-based layered oxide cathodes and the improved performance of various materials are summarized in Table 1 and Figure 1, respectively. Furthermore, ion doping, structural optimization design, and interface optimization as important modification methods have been summarized in other studies [23]. In particular, the ion doping for layered cathode materials should be studied in depth and classified by different kinds of anion/cation doping, different ratios of one ion doping, and different site doping. These methods can bring different performance enhancement effects. Especially, F is considered as a promising dopant anion for improving electrochemical properties due to its extremely high electronegativity [24].

Some previous reviews [46,47,48] have pointed out that F ion-doping contributes to the enhancement of electrochemical performance of materials. But their classification is not detailed, the explanation is not specific, and the content is not perfect enough. In addition, the role of F ion-doping and the mechanisms of performance improvement are not specifically indicated. In the case of a certain elemental composition, none of these reviews has pointed out in detail the effect on the electrochemical performance after doping with F ions. As opposed to categorizing materials from a structural point of view (P2, P3, O2), we believe analyzing materials composed of the same elements is more conducive to understanding the role of F ions. In this paper, we summarize typical F-doped layered oxide cathode materials for SIBs, and then review the modification mechanism of F-doping and the corresponding energy storage mechanism (Figure 2), including (1) increasing Na^+^ diffusion rate by widening the layer spacing; (2) enhancing lifespan by reducing the irreversible multiphase transformation and preventing the distortion collapse of crystal structure; (3) accelerating Na^+^ transport kinetics by lowering the energy barrier. The systematic summary will provide some support for the future development and application of SIBs. In addition, we make a reasonable outlook on the research of F-doped layered oxide cathode materials and the future application potential for SIBs.

## 2. F-Doped Layered Oxide Cathode Materials

The roadmap of layered metal oxide materials briefly is shown in Figure 3, including the discovery and development, the energy storage mechanism, and the structure information. With the progress of the in-depth study on it below, layered metal oxides as cathode materials for SIBs have been regarded as one of essential candidates recently.

### 2.1. Na_x_MnO_2_-F

In the 1970s, a systematic study of the structure and physical properties of A_x_MO_2_ compounds was conducted (A = Li, Na, K; M = Cr, Mn, Fe, Co) [49]. In the 1980s, the variety of MO_2_ layer stackings in A_x_MO_2_ phases was devised [55]. Specially, NaMnO_2_(NMO) had been reported earlier and are well-known for their high energy densities and safety [56].

Mn-based layered oxides have many advantages such as being inexpensive, having high special capacity, and long life span, but they always face structural collapse caused by the phase from P2 to O2 at high voltages [57]. Meanwhile, the production of Mn^3+^ is extremely prone to induce Jahn–Teller (J-T) effects [58], further leading to severe distortion and deformation of the lattice structure, reducing the reproducibility of Na^+^ (de)insertion, and affecting the electrochemical properties of the material. F ions can help to solve the above problems, and F ions can effectively improve the resistance of the cathode material and electrolyte in the high voltage range, as well as help to improve the structural integrity of the TM layer and reduce the dissolution of Mn^3+^; many research results have proven the role of F ions. Layered sodium-rich Na_1.2_Mn_0.8_O_2-y_F_y_ (y = 0–0.5) has been developed based on NMO [38], which was obtained in a basic solid-state mixing method. Under the comparison of the electrochemical performance of samples with different F contents, it was determined that Na_1.2_Mn_0.8_O_1.5_F_0.5_ has the best electrochemical performance with a specific capacity of 174 mAh g^−1^ at a current density of 10 mA g^−1^. It can also maintain a retention of 68% at a current density of 1000 mA g^−1^ after 300 cycles. However, the increase of Mn^3+^ leads to the crystal structure transformation from the O3 phase to the P2 phase, that makes the poor cycle life. In addition, as shown as in Figure 4a,b, Ti- and F-co-doped cathode oxide P2-Na_0.7_MnO_2.05_ (Ti 2%, F 8%) (NMO-0.1TF) has been prepared by adjusting different ratios of NH_4_TiF_6_/NMO [45], and it shows a very high specific capacity (227 mAh g^−1^ at 20 mA g^−1^ in the voltage range of 2.0–4.2 V and a good lifespan (96.2% after 200 cycles at 1000 mA g^−1^) based on a good structural stability. In the rate test, it has a high specific capacity of 76 mAh g^−1^ at 3000 mA g^−1^, and when the current density returned to 20 mA g^−1^, it could return to a very high specific capacity of 227 mAh g^−1^.

### 2.2. Na_x_Ni_a_Mn_1-a_O_2_-F

As we all know, F-doping can bring some performance improvement of NMO, but some drawbacks of NMO limit performance enhancement [59]. Mn^3+^ easily dissociates from the skeleton of metal oxide and dissolves into the electrolyte during Na^+^ (de)insertion, leading to structural damage [60]. In addition, the high voltage might induce irreversible phase change between P2 and O2, resulting in poor cycle stability [61]. Low average voltage affects energy density. Moreover, these problems result in a series of phase transformations and structural evolutions due to the valence changes of transition metal ions during the charging/discharging process [62]. To address these issues, transition metal ions are doped in NMO. The introduction of Ni can effectively improve the average voltage and make use of Ni^2+/3+/4+^ redox couples [63] during the electrochemical cycling. Previous reports have shown that NNMO has excellent electrochemical properties. Na_2/3_Ni_1/3_Mn_2/3_O_2_ [64] shows a specific capacity of 145 mAh g^−1^ at 0.1C and a capacity retention of 89% after 50 cycles. Compared with NMO, the structure of NNMO contains the higher number of Mn^4+^ ions; however, a large amount of Mn^4+^ without electrochemical activity will make the specific capacity decrease [65]. The introduction of Ni^2+^ will also make the specific capacity decrease and cycling stability deteriorate [66], while F ion-doping can perfectly solve the above contradictory phenomena, which can help to prepare high performance cathode materials. Thus, we review the current research progress of F-doped NaNi*_x_*Mn*_1-x_*O_2_ (NNMO) layered oxide cathode material in this section.

Na_0.6_Ni_0.05_ Mn_0.95_O_2-x_F_x_ (x = 0.00, 0.02, 0.05, 0.08) [25] has been synthesized by a simple co-precipitation method with the molar ratio of Mn:Ni = 95:5. Na_0.6_Mn_0.95_Ni_0.05_O_1.95_F_0.05_ displays a reversible specific capacity of 80.76 mAh g^−1^ at 2C with a capacity retention of 75.0% after 960 cycles and a capacity decay of 0.026% per cycle. This is performed to further optimize the ratio of Mn:Ni for better electrochemical performance and more remarkable capacity retention, especially to investigate the specific role of F^-^ in terms of structural stability. A series of Na_2/3_Ni_1/3_Mn_2/3_O_2-x_F_x_ (x = 0, 0.03, 0.05, 0.07) [32] have been synthesized by a simple high temperature solid phase reaction that exhibits excellent cycling stability. The Na_2/3_Ni_1/3_Mn_2/3_O_1.95_F_0.05_ still reaches a specific capacity of 61 mAh g^−1^ at 10C, as shown in Figure 4c. Therefore, the F-doped NNMO reached a reasonable ratio of each element, and the P2-Na_2/3_Ni_1/3_Mn_2/3_O_2_ has become one of the most promising cathode candidates for SIBs showing a high electrochemical performance due to its high theoretical capacity in a wider voltage window. Similar to the research process of NMO, the bi-ion co-doped NNMO also have been gradually investigated. P2-Na_0.67_Ni_0.33_Mn_0.67_O_2-y_F_y_ and P2-Na_0.67_Ni_0.33_Mn_0.67-x_Ti_x_O_1.9_F_0.1_ cathode materials [33] have been synthesized by a simple conventional solid-phase reaction. Among them, the P2-Na_0.67_Ni_0.33_Mn_0.37_Ti_0.3_O_1.9_F_0.1_ has a specific capacity of 128.1 mAh g^−1^at 2C in the high voltage of 2.0–4.4 V, and maintains a 77.2% retention ratio after more than 300 cycles, as shown in Figure 4d. In order to effectively reduce the electron repulsion of O^2−^-O^2−^ at low capacity and suppress the phenomenon of rapid decay due to the P2-O2 phase transition when charging above 4.2 V, the Zn/Ti/F co-doped Na_0.67_Ni_0.33_Mn_0.67_O_2_ (NNZMTOF) cathode was prepared by the sol-gel method [44], with excellent capacity retention (86% after 1000 cycles), as shown in Figure 4f. Moreover, Ca/F co-doped Na_0.67-x_Ca_x_Ni_0.33_Mn_0.67_O_2-2x_F_2x_ (x = 0, 0.01, 0.03 and 0.05) [35] has been prepared by a high temperature solid phase reaction, as shown in Figure 4e. It is worth noting that the capacity retention after 500 cycles at 1C increased from 27.1% (the undoped material) to 87.2% (the co-doped material). In order to more comprehensively show the synergistic effect of the F substitution with other cations, Min, K et al. [67] conducted a large-scale co-doped experiment and selected more than 12 transition metal cations and F^-^ to optimize NNMO, determine the optimal doping position, and examine the five co-doped ion pairs by calculating the total energy of the structure. Te-, Sb-, Hf-, Y-, and Ti-F are more favorable to enhance the structural stability of NNMO. As a result, the cations and F^-^ co-doping materials are more effective at dramatically enhancing the whole performance than the single doped cations as demonstrated via many data comparisons.

### 2.3. Na_x_(TM’)_b_(TM”)_c_Mn_1-b-c_O_2_-F (TM’, TM” = Transition Metal Elements)

Since it was significantly proven that the F-doping had greater improvement for the performance of SIBs, the iron-doped layered oxide cathodes also gradually became popular [68]. This is because the weak interaction between layers and the hollow two-dimensional space provides great convenience for the migration of Na^+^ [11]. Additionally, it suppresses the migration of other transition metal ions into the Na layer, hindering the crystal structure transformation. However, the Fe^3+^ will migrate, leading to structural instability when the Na^+^ (de)inserts [67]. So, how to solve this problem has become one of the important research directions to enhance the structural stability of iron-doped cathodes [68].

The O3-type NaNi_1/3_Fe_1/3_Mn_1/3_O_2-x_F_x_ (x = 0, 0.005, 0.01, 0.02, NNFeMO-F) cathode materials [27] were prepared by using a simple solid-phase reaction. Its capacity was 110 mAh g^−1^ at 150 mA g^−1^ and the capacity retention rate was 85% after 70 cycles, as shown in Figure 5a. Partial replacement of O by F can significantly improve the cycling performance of materials. As shown in Figure 5b, NNFeMO-F shows a good modification effect. The O3-type co-doping Mo and F increase the reversible capacity and cycling stability [29]. In addition, the P2-type Na_0.67_Ni_0.15_Fe_0.2_Mn_0.65_O_1.95_ F_0.05_ (NNFeMO-F_0.05_) [34] was firstly synthesized by the co-precipitation method, and in situ Mg substitution was performed by electrochemical method, where Mg was directly introduced into the Na site of the sodium-deficient P2 phase cathode material without occupying the TM layer. The obtained P2-type NNFeMMO-F_0.05_ has an excellent electrochemical performance with a remarkable specific capacity of 229 mAh g^−1^, as shown in Figure 5c. An appropriate F content greatly improves the cycling performance of the cathode, and the combined effect of F and Mg demonstrates excellent electrochemical performance. Even so, the effect of the F-doping still needs a more in-depth and detailed study in the future.

## 3. F-Doping Modification Mechanism

### 3.1. Increasing the Na^+^ Diffusion

The F-doped layered oxide cathode can effectively improve Na^+^ transport kinetics, and the diffusion rate of Na^+^ can be accelerated by more than 100 times if the doping ratio and location are reasonable in Figure 6a [43], as analyzed and calculated by Electrochemical Impedance Spectroscopy (EIS) curve and Constant Current Titration Technique (GITT), showing better electrochemical modification advantages. Boosting electron mobility is mainly due to the following reasons:(1) Increasing the layer spacing as in Figure 6b. With the increasing amount of F in the crystal structure, the angle of the diffraction peak (002) first slightly decreases and then increases [69]. F^-^ will have a slight effect on the local structure and cell size of the crystal, but it cannot change the overall structure of the crystal due to the fact that a small number of F^-^ replace the position of the O^2−^, and the ionic radius of the F^−^ (1.33 Å) is slightly smaller than that of the O^2−^ (1.40 Å) [34]. Due to the radius change from Mn^4+^ to Mn^3+^, the crystal changes can lead to performance changes. In detail, the doping of F^−^ makes the metal oxide layer spacing increase significantly, and the larger layer spacing will facilitate the Na^+^ (de)insertion conveniently, as shown in Figure 6c [70], which is conducive to the Na^+^ diffusion in metal oxide layers. (2) Reducing the energy barrier of the Na^+^ migration is a key factor for enhancing electrochemical performance. Therefore, studying the intensity of the energy barrier of Na^+^ migration and the effect of the F-doping are the important ways to analyze the mechanism of material modification [71].Transition state theory and density functional theory (DFT) in Figure 6d,e show that lower migration energy means faster diffusion rate. In the study of Ti- and F-co-doped NMO [45], the Na^+^ migration energy (0.80 eV) in the NMO-0.1TF crystal structure was obtained by DFT, while in the NMO crystal structure, the migration energy was as high as 1.26 eV.

### 3.2. Enhancing the Structural Stability

Due to the strong electronegativity and the single valence state of the F^−^, the cathode materials form the ultra-stable structure by the following aspects. (1) Suppressing the J-T effects. As shown in Figure 7a, the results of X-ray photoelectron spectroscopy (XPS) measurements show that the Mn^3+^/Mn^4+^ ratio increases significantly with the increase of F^-^ content due to charge balance, and it is further verified that F-doping can reduce the number of Mn^4+^, and the increased Mn^3+^ will occupy the Ni^2+^ sites and destroy the Ni^2+^/Mn^4+^ redox ion pairs, thus suppressing the J-T effects to improve structural stability [43]. (2) Increasing the disorder of transition metal sites and preventing Na^+^/vacancy ordering. The disorder induced by F-doping can break the crystal structure distortion due to J-T effects and facilitate the structural stability [32]. Based on the DFT results, in the structure of Na_0.6_Mg_0.3_Mn_0.7_O_1.95_F_0.05_, the chemical bonding between Mn and F strongly increases, and less Mn^3+^ reduces the distorted collapse of the crystal structure and structural transitions [69]. (3) Preventing the collapse of the P2 phase structure. The strong Na-F bond can enhance the excessive Na^+^ extraction and intercalation to improve the structural stability [25]. (4) Inducing partial reduction of Mn^4+^ to Mn^3+^ via charge neutrality. In Figure 7b,c, it is confirmed by electron energy loss spectroscopy (EELS) that Ni^2+^/Ni^3+^ as a pair of oxidation and reduction are mainly responsible for charge compensation during the charge/discharge process, while the redox couples of Mn^3+^/Mn^4+^ are also involved for increasing the specific capacity [32].

### 3.3. Suppress Phase Transition and Change Surface Morphology

As shown in Figure 8a, in situ X-ray diffraction (XRD) was conducted to monitor the structural change, and the cathode presented highly reversible transitions. During the Na^+^ extraction and intercalation process, the reversible P2-P4 phase transition of cathode material after doping replaced the irreversible P2-O2 phase transition in a high voltage window, exhibiting excellent structural reversibility and integrity [33]. Before Na^+^ extraction, the overall structure remains intact. After Na^+^ were removed, the thermodynamic stability of the O3-type structure usually decreased in virtue of the easy collapse of the hollow part of the structure caused by the removal of Na^+^, but Ti-F co-doping has the best effect on inhibiting phase transition and plays the most important role in structural stability and integrity [32]. Meanwhile, due to the strong electronegativity of F, the arrangement and distribution of TM elements (Ni, Fe, Mn) becomes disordered, which is favorable to suppress the structural distortion. As shown in Figure 8b, for Na_0.67_Ni_0.15_Fe_0.2_Mn_0.65_ O_1.95_F_0.05_ (NFMF-005), the fluctuation range of the lattice volume is smaller than that of Na_0.67_Ni_0.15_Fe_0.2_Mn_0.65_O_2_(NFM), indicating that the F substitution effectively suppresses the phase transition and structural distortion during the electrochemical cycling, while the plateau voltage is hysteretic, which can be explained by the fact that the F substitution modification delays the phase transition of the P2 structure, and a more stable structure has better cycling stability [34]. At the same time, doping makes it easier to form lamellar sheets, loosening the microspheres, and forming a hierarchical structure, which is conducive to accelerate the fast transport kinetics of Na^+^ [43]. In addition, when we add the amount of F^-^, the particle size increases slightly, which is consistent with earlier reports of lamellar oxides [72,73]. An appropriate amount of F-doping can improve the taper density of the layered oxide, while the difference in particle size affects the contact area between the electrode and the electrolyte, which is very important for a high reversible capacity and remarkable capacity retention. However, there are also studies that promote phase transition, as shown in Figure 8c. With F-doping, the number of Mn^3+^ increases close to that of Mn^4+^ in the study of P2-Na_1.2_Mn_0.8_O_1.5_F_0.5_ cathode material, which promotes the transformation of structure from O3 to P2 and thus facilitates Na^+^ migration kinetics [38]. Not coincidentally, as shown in Figure 8d, in the study of the modified enhancement of Na_0.44_MnO_1.93_F_0.07_ [28], the tunnel structure of the low sodium content is modified into a laminar tunnel composite structure after the F^−^ are doped into it, which has both the stability of the tunnel structure and the high reversible capacity of the laminar structure. Therefore, the layered tunnel intergrowth structure exhibits exceptional rate performance as well as outstanding cycling stability. In addition, the reaction temperature plays an important role in the formation of the material structure in this research article.

## 4. Summary and Prospects

In summary, recent progress of Mn-based transition metal layered oxide cathodes for SIBs from different aspects has been comprehensively reviewed in this paper, especially focusing on the F-doping modification. Meanwhile, this paper summarizes the significant role of F-doping in improving the diffusion rate of Na+.

Nevertheless, in the process of commercializing SIBs, there are still some difficulties that have not been effectively solved. First of all, upon Na^+^ extraction, the electrostatic attraction between the Na layers and the oxide layers disappears during the charge/discharge process, and the repulsion between the TM and the TM layers will gradually enhance. Meanwhile, the deintercalation and intercalation of the large radios of Na^+^ will leave structural voids. These phenomena seriously undermine structural stability. Secondly, for balancing the positive charge loss and maintaining charge neutrality due to Na^+^ extraction, the transition metal will undergo a redox reaction, and the structure of the cathode material will be twisted and deformed with the different transition metal valence states. Additionally, to achieve the lowest energy steady-state, the internal structure would inevitably change, resulting in the formation of Na^+^/vacancy ordering, the transition metal ions migration/dissolution, the series of phase transformations and structural evolution, and so forth. Thirdly, recent studies have also confirmed the migration of some transition metals to the sodium layer during the charging and discharging process, which will bring about many drawbacks, such as poor thermal stability, boundary side reactions, and rapid capacity degradation. Therefore, inhibiting the migration is important to reduce electrode dissolution and particle cracking.Last but not the least, at high voltages (4.2 V and above), it is often accompanied by structural irreversible phase transition, which will bring about the collapse of the overall structure and release of O_2_ gas [74]. Although significant achievements have been made in advancing SIBs in recent years, it still has a long way to go to overcome the subsistent obstacles in order to overcome the inherent problems of the material itself and put full-battery SIBs into practical application. We give an outlook on future technologies to improve electrochemical performance and safety in the following areas.

1. Dopant selection and reasonable composition design. At present, the study of F-doping mainly stays in the analysis of experimental results, and it does not clearly address the quantitative optimization of F-doping and does not address the optimal solution of the doping amount from a molecular structural perspective. Meanwhile, there are some directional studies about the selection of dopant and element, but the mechanism explanation from the atomic and molecular perspective needs to be further studied, and the mechanism of interaction between different elemental ions from the quantum perspective needs to be advanced more deeply.

2. Accurate characterization of doping sites. Guided by density functional theory calculations, the F substitution may change crystal structure and the formation energy as well as thermodynamic energy, but how to reasonably design the sites of different elements, especially F^−^, is facing great challenge. For future research on layered oxide cathodes [51], it is crucial to explore controlled and precise doping sites and provide intuitive and in-depth proof of the mechanism.

3. Multi-phase combination synergy of P2, O3, O2, O4, and other different classical types. Although the P2-type structure is conducive to the embedding and detachment of Na^+^, the P2-type layered oxide cathode usually suffers from short cycling lifespan. Meanwhile, the O3 structure is conducive to the storage of large amounts of Na^+^, but this structure is less reversible due to O3-P3 phase transitions that alter the diffusion mechanism of Na^+^ ions [75]. Our recent studies have revealed that different doping amounts of F ions affects the structure of the materials. The formation of the Na-F bond can change the bonding between O2^−^ and Na^+^, and the F ions embedded in the TM layer are conducive to enhance the electrostatic attraction between the Na layer and the TM layer, thus changing the structural morphology of the material, especially for the layered and tunneled structure, and P2, O3, O2, O4 structure. Following this line of thinking, the composite structures could successfully achieve the integration of synergistic advantages stemming from the diverse structures and great improvement in electrochemical performance. Hence, it is important to reasonably prepare the complex systems composed of different structures [76]. The biphase synergy of P2-O3, P2-P3, P2-tunnel, and other intergrowth systems will be able to make full use of the advantages of various phases to develop high-performance materials, transforming the harmful irreversible phase transition into a beneficial reversible phase transition. Some new methods could integrate the merits of multiple phases to realize performance breakthroughs.

Turning to large scale applications, there have been two schools favoring either layered or polyanionic compounds such as NaFePO_4_ [77]. Polyanionic cathode materials have been industrialized on a small scale, but polyanionic compounds inherently suffer from low stacking density [78], and it is not a perfect choice. Nowadays, the energy density of layered oxide cathode materials is similar to that of LiFePO_4_, and the trend seems to be in favor of layered oxides due to the improvement in the composition of the material, which has led to a reduction in cost. Although there are still many problems and challenges in the industrialization, low cost and safety will be the biggest advantages of SIBs [79]. Hence, we believe that the large-scale production and commercial application of high-performance cathode materials will become possible through the F-doped strategy or multiple ions co-doped strategy. In addition, close communication between academic and industrial organizations combined with best endeavors is needed to launch full-cell SIBs into practical application. As shown in Figure 9, SIBs, as low-cost batteries, are able to integrate discontinuous energy flows from renewable sources [80]. Grid storage, wind power, and solar photovoltaic power act as the main renewable energy sources for the distributed energy system (①, ②, and ③), while the main scenarios and forms of energy use in society, such as commercial electricity, residential electricity, and industrial electricity (④, ⑤, and ⑥) will promote the development of the low-cost SIBs in the field of grid energy storage systems, renewable energy conversion, and utilization. The utilization of layered oxide cathode materials serving as rechargeable SIBs at the industrial and commercial level is anticipated to commence in the foreseeable future.

## Figures and Tables

**Figure 1 molecules-28-08065-f001:**
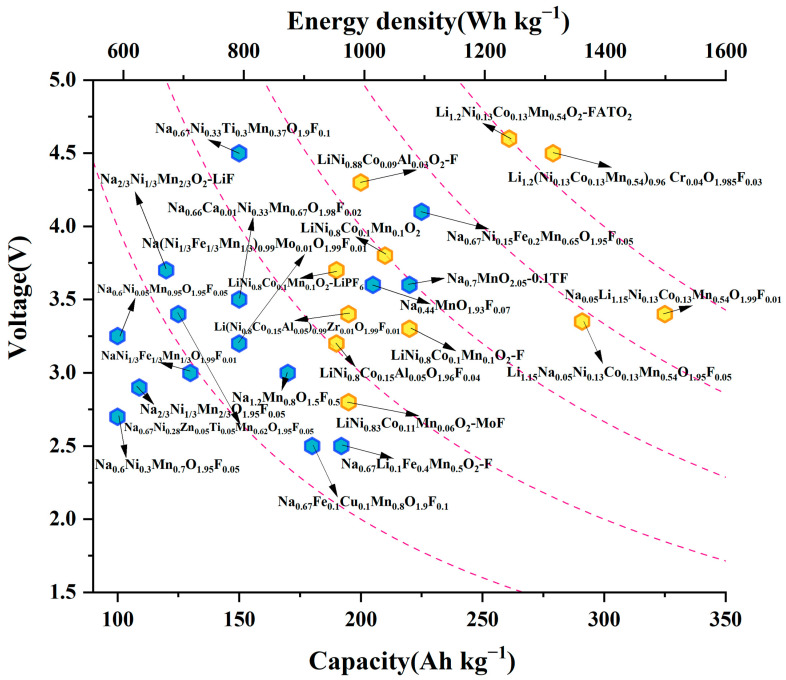
The performance comparison of the reported oxide cathode materials, including plateau potential (V vs. Na^+^/Na and V vs. Li^+^/Li for the blue and yellow hexagons, respectively) and energy density (Wh kg^−1^) [3,4,6,11,17,25,26,27,28,29,30,31,32,33,34,35,36,37,38,39,40,41,42,43,44,45].

**Figure 2 molecules-28-08065-f002:**
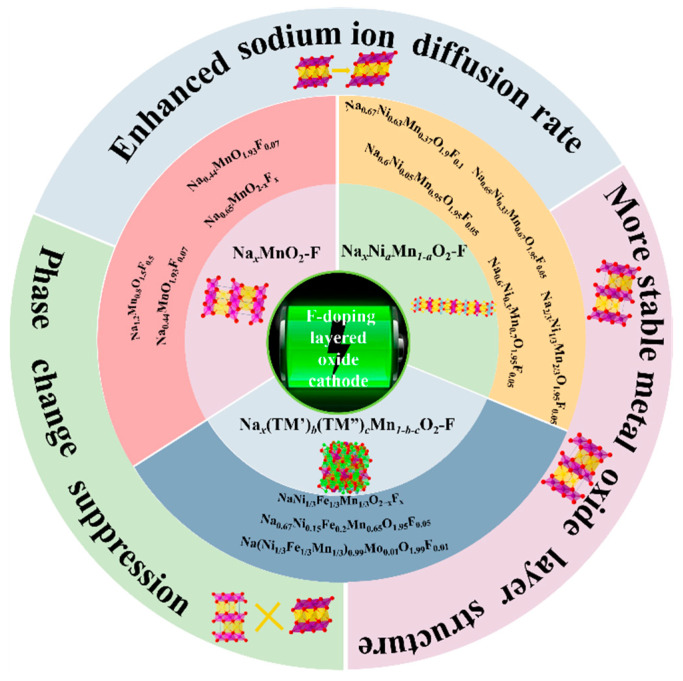
The classification of F-doped layered metal oxides: Na*_x_*MnO_2_-F, Na*_x_*Ni*_a_*Mn*_1-a_*O_2_-F, Na*_x_*(TM’)*_b_*(TM”)*_c_*Mn*_1-b-c_*O_2_-F, and the corresponding modification effects.

**Figure 3 molecules-28-08065-f003:**
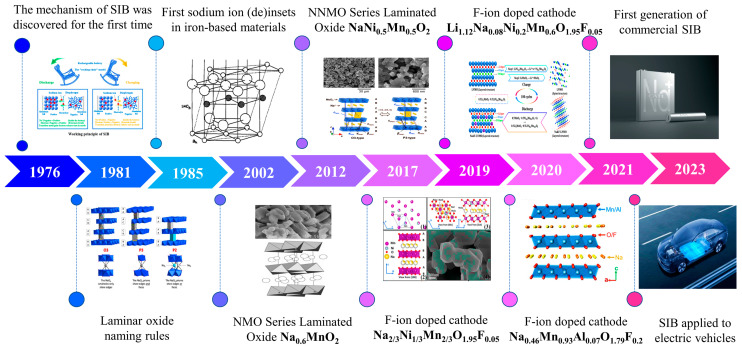
The roadmap of layered metal oxide cathode materials in SIBs [1,30,49,50,51,52,53,54]. Copyright 1991, Royal Society of Chemistry. Copyright 1994 Published by Elsevier Ltd. Copyright 2012, American Chemical Society. Copyright 2017, American Chemical Society. Copyright 2019, Elsevier Ltd. Copyright 2020, WILEY-VCH Verlag GmbH & Co. KGaA, Weinheim, Germany.

**Figure 4 molecules-28-08065-f004:**
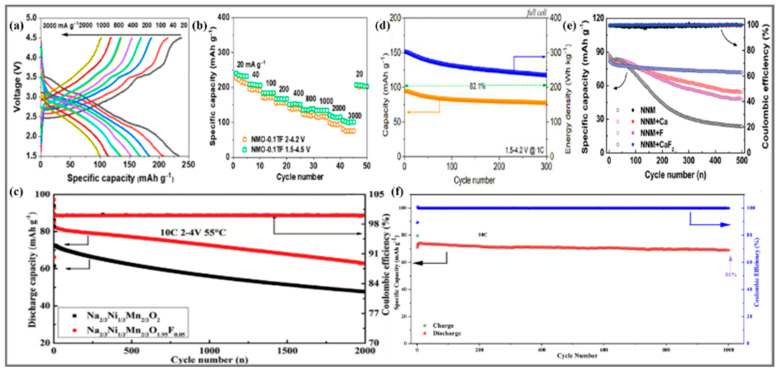
(**a**) Galvanostatic charging and discharging curves and (**b**) rate performance of NMO-0.1TF cathode at different rates in the voltage range of 1.5–4.5 V [45]. Copyright 2023, Chinese Chemical Society. (**c**) High temperature cycling performance (55 °C) of Na_2/3_Ni_1/3_Mn_2/3_O_2_ and Na_2/3_Ni_1/3_Mn_2/3_O_1.95_F_0.05_ cathode at 10 C [32]. Copyright 2020, WILEY-VCH Verlag GmbH & Co. KGaA, Weinheim, Germany. (**d**) Cycling stability of NNMTOF as cathode sodium-ion full battery [33]. Copyright 2021, Science Press and Dalian Institute of Chemical Physics, Chinese Academy of Sciences. Published by ELSEVIER B.V. and Science Press. (**e**) Comparison of cycling performance of Na_0.67_Ni_0.33_Mn_0.67_O_2_ cathodes doped with/without F at 1C in the voltage range of 2.0–4.3 V [35]. Copyright 2021, Royal Society of Chemistry. (**f**) Cycling stability of sodium-ion full batteries based on NNZMTOF and commercial hard carbon at 1C [42]. Copyright 2023, Royal Society of Chemistry.

**Figure 5 molecules-28-08065-f005:**
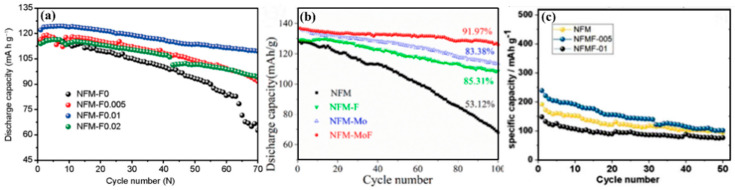
(**a**) Cycling performance of the NaNi_1/3_Fe_1/3_Mn_1/3_O_2-x_F_x_ (x = 0, 0.005, 0.01, 0.02) at 150mAg^−1^ [27]. Copyright 2017, Science China Press and Springer-Verlag GmbH Germany. (**b**) Cycling performance at 1 C for 100 cycles of all cathodes in 2.0–4.0 V [29]. Copyright 2022, Elsevier Ltd. All rights reserved. (**c**) cycle performance of P2-type Na_0.67_Ni_0.15_Fe_0.2_Mn_0.65_O_1.95_ F_0.05_ (x = 0, 0.05, 0.1) [34]. Copyright 2021, Elsevier B.V. All rights reserved.

**Figure 6 molecules-28-08065-f006:**
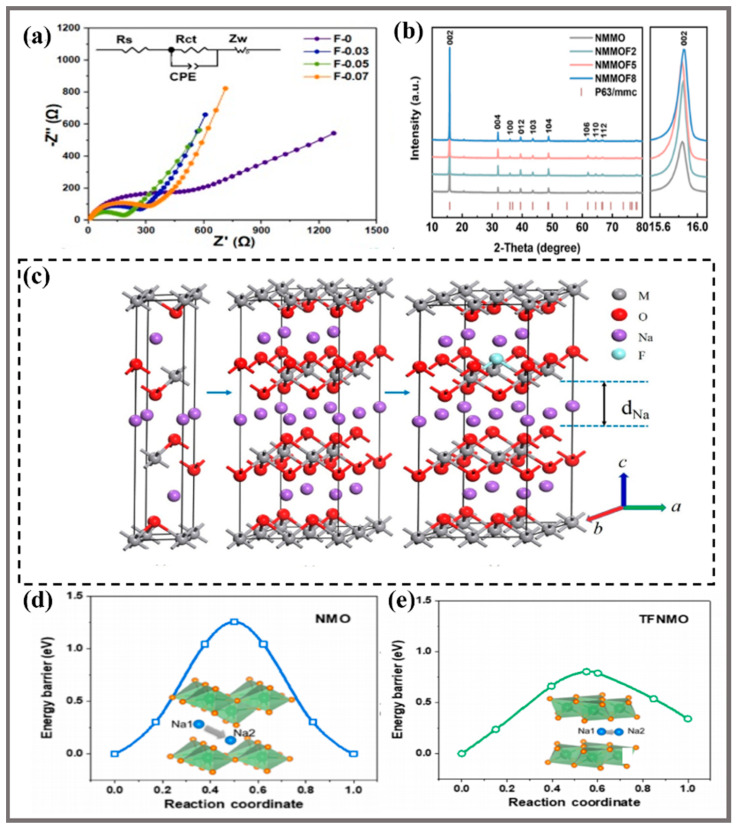
(**a**) EIS and equivalent circuits of NMO with different F-doping contents [43]. Copyright 2021, American Chemical Society. (**b**) X-ray diffraction of the NNMO material with different F-doping contents, and the detailed (002) diffraction peaks [69]. Copyright 2022, Published by Elsevier Ltd. (**c**) Schematic illustration of the supercell NMO and its F-doping [70]. (**d**,**e**) DFT calculation of Na_0.7_MnO_2.05_ (*P63*/*mmc*) and NMO (*Pmmn*) [45]. Copyright 2023, Chinese Chemical Society.

**Figure 7 molecules-28-08065-f007:**
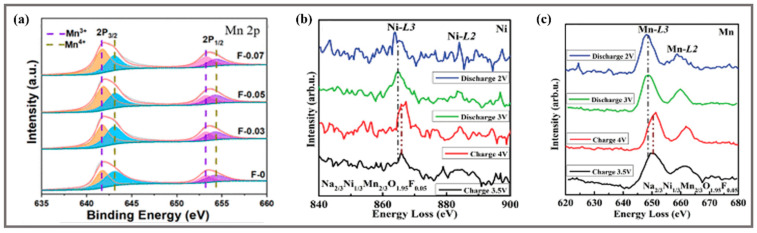
(**a**) Mn 2p high-resolution spectra of XPS patterns of the Na_0.6_Mn_0.7_Ni_0.3_O_2−x_F_x_ samples [43]. Copyright 2021, American Chemical Society. EELS spectra of Na_2/3_Ni_1/3_Mn_2/3_O_2_ [32]: (**b**) Ni L-edge signals, (**c**) Mn L-edge signals. Copyright 2020, WILEY-VCH Verlag GmbH & Co. KGaA, Weinheim, Germany.

**Figure 8 molecules-28-08065-f008:**
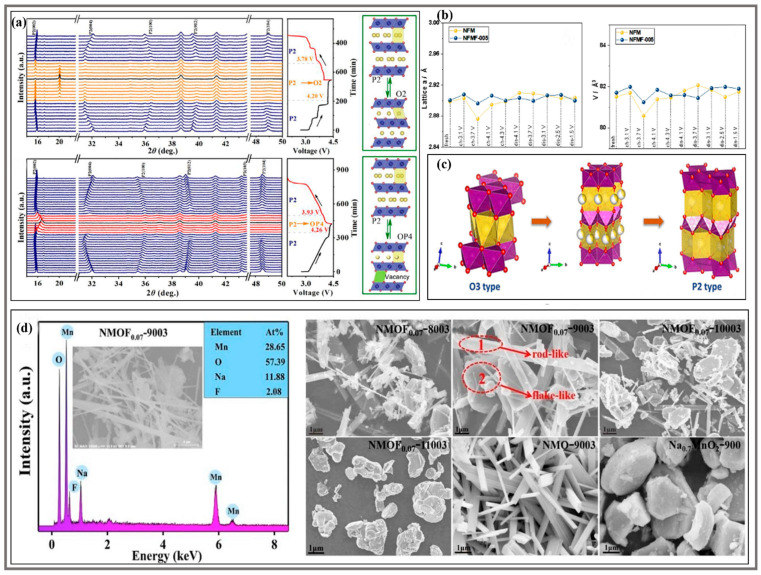
(**a**) In situ XRD patterns of Na_0.67_Ni_0.33_Mn_0.67_O_2_(NNMO) and Na_0.67_Ni_0.33_Mn_0.37_Ti_0.3_O_1.9_F_0.1_(NNMTi_0.3_OF) during the first charge/discharge process [33]. Copyright 2021, Science Press and Dalian Institute of Chemical Physics, Chinese Academy of Sciences. Published by ELSEVIER B.V. and Science Press. (**b**) Structure parameters of lattice parameter and unit cell volume of Na_0.67_Ni_0.15_Fe_0.2_Mn_0.65_O_2_ (NFM) and Na_0.67_Ni_0.15_Fe_0.2_Mn_0.65_O_1.95_F_0.05_ (NFMF-005) under different charge/discharge states [34]. Copyright 2021, Elsevier B.V. All rights reserved. (**c**) Schematic of the cathode structure evolution based on F-doping [38]. Copyright 2023, American Chemical Society. (**d**) EDS elemental mapping images of NMOF_0.07_-900 and FE-SEM images of prepared materials for NMOF_0.07_-800, NMOF_0.07_-900, NMOF_0.07_-1000, NMOF_0.07_-1100, NMO-900 and Na_0.7_MnO_2_-900 [28]. Copyright 2019, Elsevier B.V. All rights reserved.

**Figure 9 molecules-28-08065-f009:**
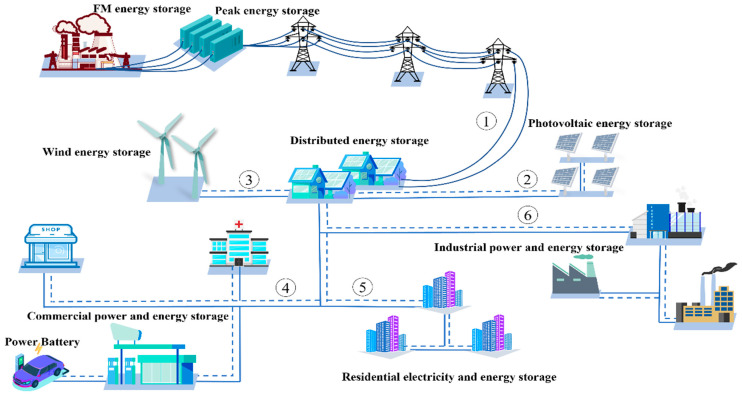
Possible scenarios for SIB in the context of new energy applications.

**Table 1 molecules-28-08065-t001:** Summary of the performances of the recent reported layered oxide cathode materials for SIBs and LIBs [3,4,6,11,17,25,26,27,28,29,30,31,32,33,34,35,36,37,38,39,40,41,42,43,44].

Materials	Synthesis Method	Cycling Performance after F-Doping	Ref.
(SIBs)			
Na_2/3_Ni_1/3_Mn_2/3_O_1.95_F_0.05_	Solid-state high temperature reaction	88%→95% (2.0–4.0 V 400 cycles at 340 mA g^−1^)	[32]
Na_0.67_Ni_0.15_Fe_0.2_Mn_0.65_O_1.95_F_0.05_	Coprecipitation method	38%→88% (1.6–4.2 V 50 cycles at 100 mA g^−1^)	[34]
Na_0.44_MnO_1.93_F_0.07_	Oxalate precursor method	36%→86% (2.0–4.0 V 150 cycles at 200 mA g^−1^)	[28]
Na_1.2_Mn_0.8_O_1.5_F_0.5_	Solid-state method	76%→90% (1.5–4.0 V 300 cycles at 1000 mA g^−1^)	[38]
Na_0.67_Ni_0.33_Ti_0.3_Mn_0.37_O_1.9_F_0.1_	Solid-state reaction	62%→82% (2.2–4.5 V 300 cycles at 170 mA g^−1^)	[33]
Na_0.6_Ni_0.3_Mn_0.7_O_1.95_F_0.05_	Solid-state reaction	65%→78% (1.5–3.8 V 900 cycles at 1000 mA g^−1^)	[43]
Na_0.6_Ni_0.05_Mn_0.95_O_1.95_F_0.05_	Co-precipitation route and solid-state reaction	71%→75% (2.5–4.0 V 960 cycles at 180 mA g^−1^)	[25]
Na(Ni_1/3_Fe_1/3_Mn_1/3_)_0.99_Mo_0.01_O_1.99_F_0.01_	Solid-state reaction	83%→92% (2.0–4.0 V 100 cycles at 130 mA g^−1^)	[29]
NaNi_1/3_Fe_1/3_Mn_1/3_O_1.99_F_0.01_	Solid-state reaction	63%→96% (2.0–4.0 V 70 cycles at 150 mA g^−1^)	[27]
Na_0.66_Ca_0.01_Ni_0.33_Mn_0.67_O_1.98_F_0.02_	Solid-state reaction	72%→94% (2.0–4.2 V 500 cycles at 200 mA g^−1^)	[35]
Na_2/3_Ni_1/3_Mn_2/3_O_2_-LiF	Sol–gel method	36%→64% (2.0–4.2 V 100 cycles at 100 mA g^−1^)	[40]
Na_0.67_Ni_0.28_Zn_0.05_Ti_0.05_Mn_0.62_O_1.95_F_0.05_	Sol–gel method	83%→90% (2.5–4.2 V 100 cycles at 170 mA g^−1^)	[44]
Na_0.67_Fe_0.1_Cu_0.1_Mn_0.8_O_1.9_F_0.1_	Solid-state reaction	52%→88% (1.5–4.5 V 200 cycles at 200 mA g^−1^)	[41]
Na_0.67_Li_0.1_Fe_0.4_Mn_0.5_O_2_-F	Co-precipitation route and solid-state reaction	19%→67% (1.5–4.0 V 200 cycles at 200 mA g^−1^)	[30]
Na_0.7_MnO_2.05_-0.1TF	Solid-state reaction	82%→96% (2.0–4.2 V 200 cycles at 1000 mA g^−1^)	[45]
(LIBs)			
LiNi_0.8_Co_0.1_Mn_0.1_O_2_-F	Solid-state reaction	62%→96% (2.8–4.2 V 100 cycles at 100 mA g^−1^)	[42]
LiNi_0.83_Co_0.1_1Mn_0.06_O_2_-MoF	Co-precipitation method	60%→80% (2.8–4.3 V 160 cycles at 200 mA g^−1^)	[36]
LiNi_0.8_Co_0.15_Al_0.05_O_1.96_F_0.04_	Co-precipitation method	72%→98% (2.8–4.3 V 100 cycles at 400 mA g^−1^)	[39]
LiNi_0.8_Co_0.1_Mn_0.1_O_2_-LiPF_6_	Solid-state reaction	77%→94% (3.0–4.3 V 100 cycles at 85 mA g^−1^)	[6]
LiNi_0.8_Co_0.1_Mn_0.1_O_2_	Solid-state reaction	45%→88% (2.8–4.3 V 200 cycles at 200 mA g^−1^)	[5]
Li_1.2_(Ni_0.13_Co_0.13_Mn_0.54_)_0.96_Cr_0.04_O_1.985_F_0.03_	Sol-gel method with citric acid as chelating agent	79%→95% (2.0–4.8 V 50 cycles at 50 mA g^−1^)	[26]
Li(Ni_0.8_Co_0.15_Al_0.05_)_0.99_Zr_0.01_O_1.99_F_0.01_	Solid-state reaction	78%→94% (2.0–4.8 V 200 cycles at 200 mA g^−1^)	[31]
LiNi_0.88_Co_0.09_Al_0.03_O_2_-F	Solid-state reaction	68%→85% (2.8–4.3 V 150 cycles at 360 mA g^−1^)	[37]
Li_1.15_Na_0.05_Ni_0.13_Co_0.13_Mn_0.54_O_1.99_F_0.01_	Carbonate co-precipitation method	89%→96% (2.5–5.0 V 100 cycles at 200 mA g^−1^)	[7]
Li_1.2_Ni_0.13_Co_0.13_Mn_0.54_O_2_-FATO_2_	Hydro/solvothermal method	79%→92% (2.0–4.8 V 500 cycles at 200 mA g^−1^)	[4]
Li_1.15_Na_0.05_Ni_0.13_Co_0.13_Mn_0.54_O_1.95_F_0.05_	Facile co-precipitation method	73%→85% (2.0–4.7 V 300 cycles at 200 mA g^−1^)	[11]

## Data Availability

All the datasets analyzed throughout the present study are available from the corresponding author on reasonable request. The data are not publicly available due to the limited content of the article.

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
