# Peer review of "Recent Advances on F-Doped Layered Transition Metal Oxides for Sodium Ion Batteries"

_molecules, 2023, doi:10.3390/molecules28248065_

Round 1

Reviewer 1 Report

Comments and Suggestions for Authors

Comments

In this review authors present the work entitled "Recent Advances on F-Doped Layered Transition Metal Oxides for Sodium Ion Batteries.” The research on new sodium transition metal layered oxides as cathode materials for sodium ion batteries and the strategies to get new structural configurations is a very relevant topic, there are many research groups working on this hot topic.  Authors have carried out an extensive search on the published works related to F-Doped materials.  Although they have done a systematic and well-defined study there are some minor issues that should be addressed before recommending the article to be published in Molecules.

 Some minor issues should be revised;

1.- The way that authors have organized all the research articles in my opinion is correct but the first part of the review has many data about different synthesis procedures, temperatures, and conditions of all the different materials. It seems to be a bit difficult and tedious to read it. Maybe combining experimental with results will help. I would also include the reasons why the improvements in the electrochemistry are those shown.

2.- I do miss some more bibliography from Prof. J.M. Tarascon, Prof. Guyomard, Prof. C. Masquelier, Prof. S. Passerini, Prof. T. Rojo... All of them leaded or currently lead research groups in Europe that are very active in layered oxide research.

3.- I would recommend the authors to homogenize the tittles and the way to number each section and subsections.

4.- I would also recommend the authors to enlarge the perspective/prospective section.

Author Response

(1). The way that authors have organized all the research articles in my opinion is correct but the first part of the review has many data about different synthesis procedures, temperatures, and conditions of all the different materials. It seems to be a bit difficult and tedious to read it. Maybe combining experimental with results will help. I would also include the reasons why the improvements in the electrochemistry are those shown.

[Suggestion] Please combine experimental with results in the first part of the review, it will be easy to read it.

Response: Thank you for your thoughtful suggestion. Combining experimental details with results makes sense, and I also see the importance of including reasons for the observed improvements in electrochemistry. I appreciate your input on the organization of the review. I have merged experimental condition and performance improvement results in electrochemistry into the table 1. Specific modifications are reflected in the highlight of table 1.

(2). I do miss some more bibliography from Prof. J.M. Tarascon, Prof. Guyomard, Prof. C. Masquelier, Prof. S. Passerini, Prof. T. Rojo... All of them leaded or currently lead research groups in Europe that are very active in layered oxide research.

[Suggestion] Please cite and learn the papers of Prof. J.M. Tarascon, Prof. Guyomard, Prof. C. Masquelier, Prof. S. Passerini, Prof. T. Rojo.

Response: Thank you for your suggestion! I appreciate your insight into the notable researchers in the field. Including more references from Prof. J.M. Tarascon, Prof. Guyomard, Prof. C. Masquelier, Prof. S. Passerini, and Prof. T. Rojo would indeed enrich the paper's content. I'll definitely look into expanding the bibliography to incorporate works from Prof. J.M. Tarascon, Prof. Guyomard, Prof. C. Masquelier, Prof. S. Passerini, and Prof. T. Rojo. Their significant contributions to layered oxide research could add valuable insights to my paper. The specific modifications are as follows:

Ref 8, 10, 60, 63, 66, 74, 75, 76, 77, 78, 79, 80

(3). I would recommend the authors to homogenize the tittles and the way to number each section and subsections.

[Suggestion] Please homogenize the tittles and the way to number each section and subsections.

Response: Thank you for your suggestion. I agree that homogenizing the titles and the numbering of sections and subsections would enhance the overall consistency of the paper. I will certainly review and make the necessary adjustments to ensure a more uniform structure. The title and number each section have been modified according to the molecules-template-2023.

(4). I would also recommend the authors to enlarge the perspective/prospective section.

[Suggestion] Please enlarge the perspective/prospective section.

Response: Thank you for your suggestion. I'm grateful for your input regarding the perspective/prospective section. I'm committed to revising and expanding that section to provide a more thorough discussion on the implications and potential avenues for further research. We have added an outlook on materials and commercialization in Section 4, which has been highlighted.

Reviewer 2 Report

Comments and Suggestions for Authors

This manuscript reviews Recent Advances on F-Doped Layered Transition Metal Oxides for Sodium Ion Batteries. The manuscript needs a major revision for publication as the following comments:

1.      The authors should mention recent reviews on Layered Transition Metal Oxides for Sodium Ion Batteries and indicate a discrepancy between this review and previously reported reviews.

2.      The structure of this manuscript gives readers an ambiguity. Section 2.1, the authors should change the title to NaxMO2 (where x and M should be italicized). Moreover, in this section, the author should discuss about advantages and disadvantages of this structure as cathode materials for SIBs. Besides, the solution to address these disadvantages and the role of F-doping need to be emphasized.

3.      Section 2.2. What is the origin of this structure? The title should be F doped NaxNiaMn1-aO2 for this section. Moreover, in this section, the author should discuss about advantages and disadvantages of this structure as cathode materials for SIBs. Besides, solution to overcome these disadvantages and the role of F-doping need to be emphasized.

4.      Section 2.3: What is the origin of this structure AxPbNaM1-a-bO2? Please use another letter instead of Na because it is the same as sodium (Na).

5.      How about single F-doping and co-doping?

6.      The authors should give their own comments instead of reporting previous works

Comments on the Quality of English Language

Line 135

Author Response

(1). The authors should mention recent reviews on Layered Transition Metal Oxides for Sodium Ion Batteries and indicate a discrepancy between this review and previously reported reviews.

[Suggestion] Please mention recent reviews and indicate a discrepancy.

Response: Thank you for your valuable suggestion. I appreciate your suggestion to include recent reviews on Layered Transition Metal Oxides for Sodium Ion Batteries and to indicate a discrepancy between this review and previously reported reviews. I will carefully consider this advice and make the necessary revisions to enhance the comprehensiveness and accuracy of the paper. The specific modifications are as follows:

Some previous reviews have pointed out that F-ion doping contributes to the enhancement of electrochemical performance of materials. But, their classification is not detailed, the explanation is not specific, the content is not perfect enough. In addition, the role of F-ion doping and the mechanism of performance improvement are not specifically indicated. In the case of a certain elemental composition, none of these reviews has pointed out in detail the effect on the electrochemical performance after doping with F ions. We believe analyzing materials composed of the same elements is more conducive to understanding the role by F ions. In this paper, we summarize typical F- doped layered oxide cathode materials for SIBs, and then review the modification mechanism of F- doping and the corresponding energy storage mechanism (Figure 2), including (1) Increasing Na+ diffusion rate by widening the layer spacing; (2) Enhancing lifespan by reducing the irreversible multiphase transformation and preventing the distortion collapse of crystal structure; (3) Accelerating Na+ transport kinetics by lowering the energy barrier. The systematic summary will provide some supports for the future development and application of SIBs. In addition, we make a reasonable outlook on the research of F- doped layered oxide cathode materials and the future application potential for SIBs.

(2). The structure of this manuscript gives readers an ambiguity. Section 2.1, the authors should change the title to NaxMO2 (where x and M should be italicized). Moreover, in this section, the author should discuss about advantages and disadvantages of this structure as cathode materials for SIBs. Besides, the solution to address these disadvantages and the role of F-doping need to be emphasized.

[Suggestion] Please change the title to NaxMO2, discuss about advantages and disadvantages of this structure, and the role of F-doping need to be emphasized.

Response: Thank you for your thoughtful suggestion. I appreciate your suggestions to modify the title to NaxMO2 with italicized x and M. Additionally, following your suggestions we have emphasized NaxMO2 cathode materials have some inherent advantages and disadvantages. Then, we have pointed out that the doping of F ions in NaxMO2 can play a role in improving the electrochemical performance, and the experimental data of previous researchers proves the role. Nevertheless, the more specific role in the micromolecular perspective will be explained in Section 3. The specific modifications are as follows:

Mn-based layered oxides have many advantages such as inexpensive, high special capacity, and long life span, but they always face structural collapse caused by the phase from P2 to O2 at high voltages [52]. Meanwhile, the production of Mn3+ is extremely prone to induce Jahn-Teller (J-T) effects [53], further leading to severe distortion and deformation of the lattice structure, reducing the reproducibility of Na+ (de)insertion, and affecting the electrochemical properties of the material. F ions can help to solve the above problems, and F ions can effectively improve the resistance of the cathode material and electrolyte in the high voltage range, as well as help to improve the structural integrity of the TM layer and reduce the dissolution of Mn3+. Many research results have proved the role of F ions.Layered sodium-rich Na1.2Mn0.8O2-yFy (y=0-0.5) has been developed based on NMO [36], which was obtained on a basic solid-state mixing method.

(3). Section 2.2. What is the origin of this structure? The title should be F doped NaxNiaMn1-aO2 for this section. Moreover, in this section, the author should discuss about advantages and disadvantages of this structure as cathode materials for SIBs. Besides, solution to overcome these disadvantages and the role of F-doping need to be emphasized.

[Suggestion] Please change the title to NaxNiaMn1-aO2, discuss about advantages and disadvantages of this structure, and the role of F-doping need to be emphasized.

Response: Thank you for your detailed suggestion. I appreciate your suggestion to provide clarity on the origin of the structure in our study. About the origin of this structure, we learned that most researchers have conducted extensive and in-depth studies on this material (NaxNiaMn1-aO2), so we chose this material as the main object of study to explore the specific role played by F ion doping. At the same time, we also point out the advantages and disadvantages of this material and emphasize the methods of overcoming the disadvantages in this part, as well as the role of doped F ions. The specific modifications are as follows:

As we all know, F- doping can bring some performance improvement of NMO, but some own drawbacks of NMO limit performance enhancement [55]. Mn3+ easily dissociate from the skeleton of metal oxide and dissolve into the electrolyte during Na+ (de)insertion, leading to structural damage [56]. And the high voltage might induce irreversible phase change between P2 and O2 resulting in the poor cycle stability[58]. Low average voltage affects energy density. Moreover, these problems result in a series of phase transformations and structural evolution with the valence changes of transition metal ions during the charging/discharging process[59]. To address these issues, transition metal ions are doped in NMO. The introduction of Ni can effectively improve the average voltage and make use of Ni2+/3+/4+ redox couples [79]. during the electrochemical cycling. Previous reports have shown that NNMO has excellent electrochemical properties, Na2/3Ni1/3Mn2/3O2 [60] shows a specific capacity of 145 mAh g-1 at 0.1C and a capacity retention of 89% after 50 cycles. Compared with NMO, the structure of NNMO contains the more number of Mn4+ ions, but a large amount of Mn4+ without electrochemical activity will make the specific capacity decrease [61]. The introduction of Ni2+ will also make the specific capacity decrease and cycling stability deterioration [75], while F ion doping can perfectly solve the above contradictory phenomena, which can help to prepare high performance cathode materials. Thus, we review the current research progress of F- doped NaNixMn1-xO2 (NNMO) layered oxide cathode material in this section.

(4). Section 2.3: What is the origin of this structure AxPbNaM1-a-bO2? Please use another letter instead of Na because it is the same as sodium (Na).

[Suggestion] Please change the title.

Response: Thank you for your thoughtful suggestion. I appreciate your attention to detail. I have used a different letter to represent the element in the revised manuscript to avoid any ambiguity. The structure AxPbNaM1-a-bO2 has been replaced by NaxPbNcMn1-b-cO2 in Section 2.3. The structure NaxPbNcMn1-b-cO2 originates from summarizing research findings. After analyzing the specific role of F ions in NaxNiaMn1-aO2, we have found that several other researchers have studied other structure (TM layer contains three transition metal elements), such as NaNi1/3Fe1/3Mn1/3O2-xFx, Na0.67Ni0.15Fe0.2Mn0.65O1.95F0.05 et all.

(5). How about single F-doping and co-doping?

[Suggestion] Please explain the single F-doping and co-doping.

Response: Thanks for your question. Single F-doping and co-doping are defined based on the NaxMO2 structure. If the cathode material's transition metal layers consist of two or more types of transition metal elements and fluorine (F), we define the material is the co-doping cathode material.

(6).The authors should give their own comments instead of reporting previous works.

[Suggestion] Please give their own comments instead of reporting previous works.

Response: Thanks to your suggestion. As shown in the highlighted parts of the article (such as P5, 7, 8, 9, 14, 15, 16) we have added many of our own comments and removed the simple electrochemical data in the Section 2, as well as avoided reporting previous works. These changes will make the article easier to read.

Round 2

Reviewer 2 Report

Comments and Suggestions for Authors

1. "Some of the previous review have pointed out that...". Please cites these reviews.

2.  NaxPbNcMn1-b-cO(P, N = doped metal). Please use a different letter for P and N as they are the same as Phosporus and Nitrogen.

Comments on the Quality of English Language

Minor editing of English language required

Author Response

Dear Editor and Reviewers,

Thanks very much for your letter and the Reviewers’ comments concerning our manuscript entitled “Recent Advances on F-Doped Layered Transition Metal Oxides for Sodium Ion Batteries.” Those new comments are valuable and very helpful for revising and improving our paper. The main corrections in the paper and the responses to the reviewers’ comments are given below:

 # Referee:

Comments:

(1)."Some of the previous reviews have pointed out that...". Please cites these reviews.

[Suggestion] Please cite the previous reviews.

Response: Thank you for your thoughtful suggestion. I will certainly consider integrating these references into the revised version. Your advice is valuable in enhancing the credibility and depth of the content. Specific modifications are reflected in the highlight of Section 1.

Some previous reviews[46–48] have pointed out that F-ion doping contributes to the enhancement of electrochemical performance of materials. But their classification is not detailed, the explanation is not specific, the content is not perfect enough. In addition, the role of F-ion doping and the mechanism of performance improvement are not specifically indicated. In the case of a certain elemental composition, none of these reviews has pointed out in detail the effect on the electrochemical performance after doping with F ions. As opposed to categorizing materials from a structural point of view (P2, P3, O2), we believe analyzing materials composed of the same elements is more conducive to understanding the role of F ions.

Ref 46,47,48

(2). NaxPbNcMn1-b-cO2 (P, N = doped metal). Please use a different letter for P and N as they are the same as Phosporus and Nitrogen.

[Suggestion] Please replace Please replace the letters P and N.

Response: Thank you for your valuable suggestion regarding the chemical formula representation. I appreciate your keen observation. To address the potential confusion with Phosphorus and Nitrogen, I will make the necessary adjustments and use distinct letters for the doped elements P and N. This modification will enhance the clarity of the formula in the paper. The specific modifications are as follows:

Nax(TM’)b(TM’’)cMn1-b-cO2-F (TM’, TM’’= transition metal elements)